# Sentiment Computation of UK-Originated COVID-19 Vaccine Tweets: A Chronological Analysis and News Effect

Olasoji Amujo [1], Ebuka Ibeke [1,*], Richard Fuzi [1], Ugochukwu Ogara [1,2] and Celestine Iwendi [3,4,*]

1  School of Creative and Cultural Business, Robert Gordon University, Aberdeen AB10 7AQ, UK
2  Aberdeen & Grampian Chamber of Commerce, Aberdeen AB23 8GX, UK
3  School of Creative Technologies, University of Bolton, A676 Deane Rd., Bolton BL3 5AB, UK
4  Department of Mathematics and Computer Science, Coal City University Enugu, Enugu 400231, Nigeria
*  Correspondence: e.ibeke@rgu.ac.uk (E.I.); celestine.iwendi@ieee.org (C.I.)

**Abstract:** This study aimed to analyse public sentiments of UK-originated tweets related to COVID-19 vaccines, and it applied six chronological time periods, between January and December 2021. The dates were related to six *BBC* news reports about the most significant developments in the three main vaccines that were being administered in the UK at the time: Pfizer-BioNTech, Moderna, and Oxford-AstraZeneca. Each time period spanned seven days, starting from the day of the news report. The study employed the bidirectional encoder representations from transformers (BERT) model to analyse the sentiments in 4172 extracted tweets. The BERT model adopts the transformer architecture and uses masked language and next sentence prediction models. The results showed that the overall sentiments for all three vaccines were negative across all six periods, with Moderna having the least negative tweets and the highest percentage of positive tweets overall while AstraZeneca attracted the most negative tweets. However, for all the considered time periods, Period 3 (23–29 May 2021) received the least negative and the most positive tweets, following the related *BBC* report—'COVID: Pfizer and AstraZeneca jabs work against Indian variant'—despite reports of blood clots associated with AstraZeneca during the same time period. Time periods 5 and 6 had no breaking news related to COVID vaccines, and they reflected no significant changes. We, therefore, concluded that the *BBC* news reports on COVID vaccines significantly impacted public sentiments regarding the COVID-19 vaccines.

**Keywords:** COVID-19; sentiment analysis; tweets; breaking news; vaccines

## 1. Introduction

The novel coronavirus, SARS-CoV-2, which caused the coronavirus disease COVID-19, was first detected in Wuhan, Hubei Province, the People's Republic of China, in December 2019 and was declared a pandemic by the World Health Organisation (WHO) on 11 March 2020. By 27 October 2021, there had been more than 4.9 million COVID-19-related deaths and 245.5 million confirmed COVID-19 cases reported globally [1]. As of 27 October 2021, more than 6.94 billion doses of COVID-19 vaccines had been administered worldwide, which enabled at least 48.9% of the total global population to be at least partially vaccinated and 37.9% were fully vaccinated against COVID-19 [2] While 272 vaccines were in development against COVID-19, with 104 vaccines in clinical testing and 22 in use as of 29 October 2021 [3], it was the Pfizer-BioNTech, Moderna, and Oxford-AstraZeneca COVID-19 vaccines that were of interest to this study, owing to their crucial role in the UK's mass immunisation programme [4]. The pandemic disrupted activities all over the world, as schools and education systems were halted and eventually proceeded to online studies [5,6]. Another worrying development of the pandemic was information overload, which included misinformation and confusing messages (e.g., 'fake news') [7], leading to contrasting opinions among the general public [8,9].

### 1.1. The COVID-19 Pandemic and Vaccinations in the UK

The first officially confirmed COVID-19 case and death in the United Kingdom (UK) were discovered in January and March, respectively. The UK was the first western country to authorise a COVID-19 vaccine for emergency use with the approval of the Medicines and Healthcare products Regulatory Agency (MHRA) on 2 December 2020 and, subsequently, became the first country to start mass immunisation with the Pfizer-BioNTech COVID-19 vaccine on 3 December 2020. The Oxford-AstraZeneca vaccine was approved in the UK by the MHRA on 30 December 2020 and was administered outside of clinical trials on 4 January 2021. The Moderna COVID-19 vaccine was first administered on 7 April 2021. Promoting COVID-19 vaccination uptake was necessary for reducing the burden on the National Health Service (NHS), especially since the number of patients in UK hospitals with confirmed COVID-19 infections had been increasing. The number of patients in mechanical ventilation (MV) beds in UK hospitals had also increased significantly. Furthermore, the number of daily deaths within 28 days of testing positive for COVID-19 had increased approximately 20-fold in the UK within 5 months: from 10 deaths on 27 May 2021 to 207 deaths on 27 October 2021. By 27 October 2021, there had been more than 140,000 confirmed deaths within 28 days of testing positive for COVID-19 and 8.8 million confirmed COVID-19 cases reported. However, as of 27 October 2021, there had also been more than 102.1 million doses of COVID-19 vaccines administered in the UK, with more than 49.7 million recipients receiving their first COVID-19 vaccine dose [10]. Moreover, the COVID-19 booster vaccination programme was already underway by 27 October 2021 for people most at risk of COVID-19 (without the third dose of vaccine); thus, promoting vaccine uptake continued to be indispensable for protecting people against severe illness [11].

### 1.2. COVID-19 Vaccine Sentiment on Twitter

Government-imposed public health interventions, such as stay-at-home orders and quarantines, introduced additional challenges for the traditional methods used for gathering data on people's opinions [12]. For instance, utilising surveys for data collection had drawbacks even before the COVID-19 pandemic, manifesting in their small and unrepresentative samples or relatively expensive and time-consuming implementation [13,14]. Therefore, researchers such as Hussain et al. [15] advocated using social media data to examine public sentiments due to the possibility of acquiring more representative samples with the right spatiotemporal granularity and often at a lower cost than surveys. Social media was a suitable data source for this project as around 82% of adults that use the internet in the UK have a profile on social media with roughly consistent representation from all socio-economic groups and genders [16]. One of the largest social networking platforms is Twitter, with over 211 million daily active users around the world, as of the third quarter of 2021, which made it a feasible data source for conducting sentiment analysis on COVID-19 vaccine-related discourse. Moreover, Twitter users have, thus far, been willing to express their views openly on this platform due to their perceptions of anonymity [17] and the lack of stringent scientific vetting or editorial curation of tweets on the platform [18]. Consequently, Twitter can be a valuable data source for researching public sentiment towards COVID-19 vaccines. Legitimate news events reported by authoritative news organisations worldwide could potentially influence the sentiment towards COVID-19 vaccines among Twitter users [19]. This project analysed the sentiment of UK-originated tweets, and since the *British Broadcasting Corporation (BBC)* continues to be the news organisation with the highest reach among UK-based adults, the *BBC* was regarded as the most influential authoritative news organisation in the UK.

### 1.3. The Benefit of Understanding COVID-19 Vaccine Sentiments

Sentiment analysis of UK-originated tweets about specific COVID-19 vaccines was one possible avenue for understanding public sentiment towards COVID-19 vaccines in the UK. As sentiment analysis could be conducted retrospectively, this project focused on the days following the publication of certain *BBC News* reports. These news articles were selected based on their perceived potential to generate positive or negative sentiment towards

the three COVID-19 vaccines used in the UK's vaccination programme. By exploring the general sentiment about COVID-19 vaccines at these times by examining the likely effects of the news reports on the public sentiment towards COVID-19 vaccines and comparing the sentiment towards different COVID-19 vaccines over time, this project could provide useful insights for the UK authorities.

The contributions of this paper were 4-fold:

- To explore the general sentiment about COVID-19 vaccines in the UK
- To chronologically examine the potential effects of news reports on the public sentiment towards COVID-19 vaccines
- To compare the sentiments toward different COVID-19 vaccines as related to these news reports
- To compare the sentiment towards COVID-19 vaccines over time

Many studies have utilized sentiment analysis to research public opinion on COVID-19 vaccines, especially on the effect of media on these opinions. However, there are still some research gaps that this study seeks to fill. To the best of our knowledge, no existing study performed a chronological analysis to understand how news reports impact people's sentiments. Particularly, there was a lack of research on understanding public sentiment towards COVID-19 vaccines, specifically in the UK context. This study aimed to address this gap by analyzing tweets originating from the UK to understand public sentiment towards COVID-19 vaccines over time. Furthermore, there was limited research on the impact of news and events on public sentiment towards vaccines. This study aimed to address this gap by chronologically analyzing tweets in relation to specific news and events to understand how they impacted public sentiment towards COVID-19 vaccines and how this varied over time. It was common knowledge that there were positive outcomes associated with news reports about the various vaccines, but there were also troubling negative outcomes, especially with the emergence of variants of the virus. Therefore, the main objective of this research was to explore the influence on people's sentiments that was exerted by different news reports about COVID-19 vaccines chronologically over one year. The research findings provided valuable insights for policymakers and practitioners by providing an understanding of public sentiment towards COVID-19 vaccines, the impact of news and events on public sentiment, the extent of misinformation about vaccines, and directions for developing effective communication strategies. The findings can help policymakers and practitioners to develop more effective strategies to increase vaccine uptake, counter misinformation, and protect public health.

## 2. Literature Review

The COVID-19 pandemic generated a significant amount of research around the world. Sentiment, contents, and retweets: A study of two vaccine-related Twitter datasets, published by Blankenship et al. [20] in 2018, revealed that leveraging the support of key thought leaders on social media to promote health education on vaccinations via tweets could reach a bigger audience. The online research focused on how users interacted with the tweets. The research by Deiner et al. [21] found that the public's interest and attitude toward vaccine hesitancy could be observed in social media conversations. They suggested that users were more likely to interact with negative news about vaccines. The reason for vaccine scepticism, as suggested in [22], could be due to how the government or the manufacturer of a specific vaccine communicated the vaccine's benefits. Using semantic network analysis, the study aimed at raising the trust of vaccines amongst a growing vaccine-hesitant population, and it indicated effective approaches to better instruct the target audience. The study suggested that the government should also address concerns that may contribute to vaccine hesitancy. However, despite the fact that the above studies were effective in establishing how public health communication and social media aid people's scepticism, they failed to address the strongest influences on people's opinions.

During health crises, such as the COVID-19 pandemic, newspapers are indispensable. They inform citizens about events around them and how they impact their lives. The

authors of Gesualdo et al. [23] posed the question "How do Twitter users react to TV broadcasts dedicated to vaccines in Italy?" The study revealed that social media discussions in response to public news were highly insightful and could be used to inform vaccine promotion communication strategies. The study advised that the implementation of a mechanism for monitoring the public's mood toward vaccines on social media should be considered by public health organizations.

Using a logistic regression model on sample data of over one million tweets, Chen and Dredze [24] analysed how images on vaccine-related tweets correlated with the likelihood of the post being retweeted, and they found that posts with images were twice as likely to be retweeted and shared than posts without an image. They also recommended that the use of images should be factored into the communication strategies for vaccine administration. While this study was carried out before the COVID-19 outbreak, it highlighted the impact of communication strategies on public sentiment regarding vaccines in general. Another study by Nezhad and Deihimi [25] analyzed Persian tweets to understand the Iranian people's view towards domestic- versus foreign-manufactured COVID-19 vaccines. Using a dataset of over 800,000 tweets posted between April and September 2021, they applied a deep-learning sentiment analysis model based on Convolutional Neural Network (CNN) and Long Short-Term Memory (LSTM) network (CNN-LSTM) architecture. They found a subtle difference in the Persians' views towards domestic versus imported vaccines: Foreign vaccines had higher positive perceptions. While this study clearly underscored the need for positive promotion of vaccines on social media, it did not account for the cultural factors that could impact the overall perceptions and conversations of Persians about the topic of vaccine administration in Iran.

Another COVID-19 sentiment analysis was carried out by Yan et al. [26] on a different social media platform, Reddit, across different cities in Canada and found three main discussion categories based on topics including vaccines, vaccine uptake, and vaccine supply. The level of discussion within these topics correlated positively with vaccine acceptance in Canada. This study, similar to [24,25], showed that social media could be used to better understand sentiments about COVID-19 vaccines and potentially help improve communication about vaccines. This finding was in line with the study by Jang et al. [27], which tracked the attitudes of Twitter users in Canada following vaccine roll-outs. In the study, Jang et al. [27] identified two groups of Twitter users who harnessed negative sentiments to achieve divergent goals; the 'anti-vaxxers', who used negative sentiments to discourage vaccine acceptance, and the 'COVID zero' group, who used negative sentiments to criticise the public health response while encouraging vaccination. This is additional evidence of the impact of social media communication on COVID-19 vaccine participation. It was also discovered in a study by Monselise et al. [28] that fear was among the top-5 emotions identified in nearly 8 million tweets collected during a 60-day window that started on 16 December 2020. This study was conducted in the United States. In addition, some sentiments could have been influenced by election misinformation following the U.S. presidential election on 4 November 2020 [29].

Furthermore, nearly a million and a half unique tweets from over half a million Twitter users were collected by Lyu et al. [19] between March 2020 and January 2021 to understand the changes in public concerns and how they impacted the goal of achieving herd immunity. They found that trust was the most predominant emotion during the selected time period and reached its peak in November 2020, following the announcement of the 90% efficacy of the Pfizer vaccine [30]. This study provided a snapshot of a time during the COVID-19 pandemic when people were, for the first time, required to perform physical and social distancing, isolation, and quarantine, which also had psychological impacts [31]. This indicated that desperation, fear, and hope could have played significant roles in driving public sentiments at the time. According to the study in [32], people's attitudes and emotions towards COVID-19 evolved over time, from positive (at the beginning of the pandemic) to negative towards the end of 2021. This was further supported by the study carried out by Niu et al. [33], who attempted to understand the reason for the rapid

acceptance of COVID-19 vaccination by the Japanese population at a time when global confidence in vaccines was diminishing (January to September 2021), based on Twitter comments. They found that negative sentiments outweighed the positive ones, with no increased vaccine confidence, but the communication strategy adopted by the public health authorities generated awareness about the danger of COVID-19, which was key to increasing vaccine uptake.

### 3. Methodology

As shown in Figure 1, this study focused on the extraction and sentiment analysis of tweets (restricted to tweets in the English language that were generated in the UK) relating to COVID-19 vaccines over a period of time. The time periods were selected according to *BBC News* reports and breaking news about the vaccines. Therefore, the methodological process of this research began by identifying periods during the pandemic when *BBC News* broadcasted significant news stories relating to any of the three vaccines administered in the UK: Pfizer-BioNTech, Moderna, and Oxford-AstraZeneca. This was then followed by the extraction of tweets that mentioned any of these vaccines during the selected time periods.

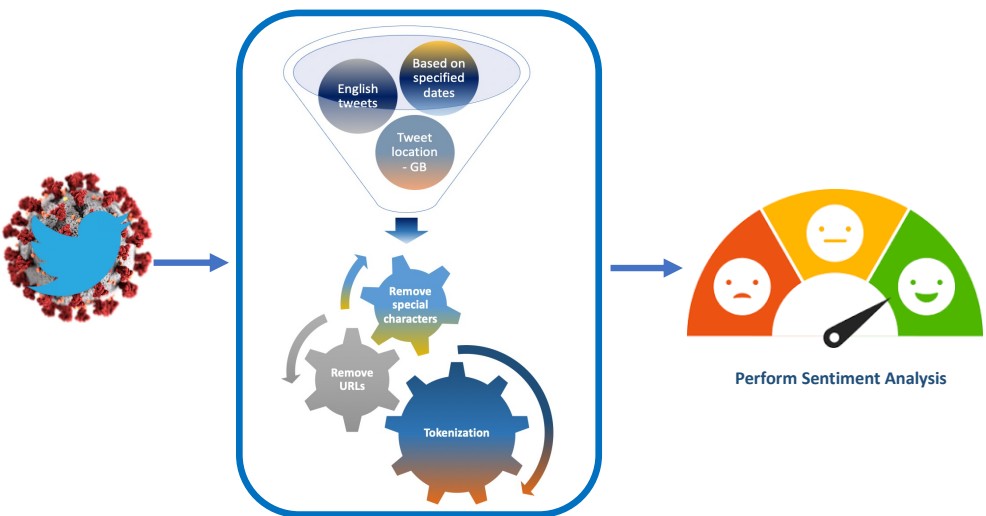

**Figure 1.** Model architecture.

### 3.1. Dataset

Six data periods were considered in this research, and they are based on the *BBC News*' reportage of significant developments regarding the UK's COVID-19 vaccines. Each period spanned seven days, starting from the day of the news report. The time periods and the news headlines are shown in Table 1. The search terms for each of the vaccines are displayed in Table 2. The table shows the terms relating to the three vaccines. As shown in the table, some vaccines have more search terms than others. To the best of our knowledge, these were valid combinations of terms used to refer to the vaccines.

Figure 2 shows the number of tweets extracted over each time period for the three vaccines. We observed that the number of tweets for AstraZeneca was highest in the first period and consistently decreased. In reverse, Moderna had the lowest number of tweets during the first time period but increased consistently over the time periods. Pfizer appeared to have the most tweets between January and December 2021. We observed that from the third to the sixth periods, it had the highest number of tweets.

**Table 1.** Tweet extraction periods and news titles.

| Period | News |
|---|---|
| 25 January–31 January 2021 | Moderna vaccine appears to work against variants |
| 2 April–8 April 2021 | COVID: 30 blood clot cases found in AstraZeneca recipients in the UK |
| 23 May–29 May 2021 | COVID: Pfizer and AstraZeneca jabs work against Indian variant |
| 9 July–15 July 2021 | Heart inflammation link to Pfizer and Moderna jabs |
| 26 November–2 December 2021 | New COVID variant (Omicron): Javid says UK must act quickly over public health risk |
| 8 December–14 December 2021 | COVID: Vaccines should work against Omicron variant, WHO says |

**Table 2.** Twitter search terms.

| Vaccine | Search Terms |
|---|---|
| Moderna | (modernavaccine OR modernajab OR modernaCOVIDvaccine OR moderna-COVID19vaccine OR moderna) |
| AstraZeneca | (AstraZenecavaccine OR AstraZenecajab OR oxfordvaccine OR oxfordjab OR oxfordCOVIDvaccine OR oxfordCOVID19vaccine OR AstraZenecaCOVIDvaccine OR AstraZenecaCOVID19vaccine OR AstraZenecaoxfordvaccine OR oxfordAstraZenecavaccine OR AstraZeneca) |
| Pfizer | (pfizervaccine OR pfizerjab OR biontechvaccine OR pfizerbiontechvaccine OR biontechpfizervaccine OR pfizerCOVIDvaccine OR pfizerCOVID19vaccine OR pfizer) |

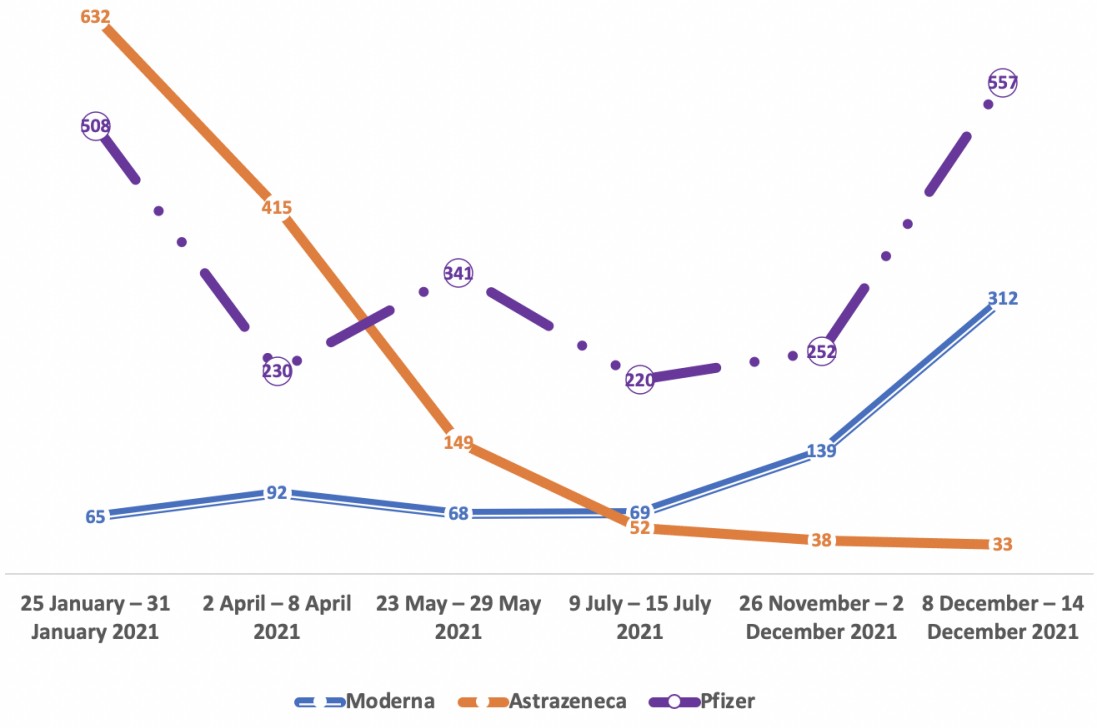

**Figure 2.** Number of tweets extracted over time.

### 3.2. Data Preprocessing

While extracting the tweets, we established initial restrictions and then pre-processed the data, as follows:

1.  We restricted the tweets to original tweets posted in English and generated in the UK within the time frames, as per Table 1; retweets were discarded.
2.  Hashtags, usernames, and hyperlinks were removed.
3.  Special characters were filtered and white spaces were removed.
4.  The data were tokenised before being input into the BERT model.

### 3.3. Sentiment Detection

In this study, we employed the bidirectional encoder representations from the transformers (BERT) model to analyse the sentiments in tweets [34]. The BERT model was pre-trained for deep bidirectional representation using unlabeled texts. It considered the contexts of the input data by learning from both the left–right and right–left sequences of the texts. Due to this bidirectional representation of the model, stop-word removal, stemming, and lemmatisation were not useful, as these would affect the constructs of the text data. For example, if we applied the stop-word removal to the text, *the dinner was not good*, then the result would be, *dinner good*. The sentiment would then be construed as positive. However, the negation 'not' is required to determine the negative sentiment in the text. The BERT model was trained on a large corpus of text data from the internet, including books, articles, and websites. The model was trained to predict masked words based on the context of the surrounding text using a masked language modelling task. The training corpus consists of over 3 billion tokens, and the model is designed to capture both the context of words to the left and right of a target word. The model adopted the transformer architecture and used the following two strategies during training:

1.  Masked language model
2.  Next sentence prediction

The BERT model in Figure 3 presents these strategies. Before word sequences were fed into BERT, a portion of the words (15%) was replaced with a masked token. Furthermore, based on the contexts of the other non-masked words in the sequence, the model attempted to predict the masked words. To predict the next sentence (i.e., according to the connections between the first and second sentences), the entire input sequence was passed through the transformer model. The first sentence was the classification sentence (CLS); all the sentences were separated by the separation token (SEP). During training, the second sentence came after the first 50% of the time, and for the half, a randomly sampled sentence followed. The BERT model then determined if the second sentence was a correct sequence or a random sentence.

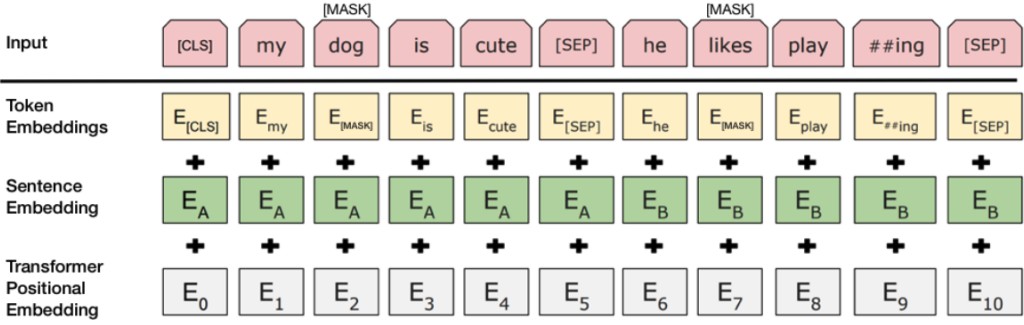

**Figure 3.** Bert model process [35].

As earlier stated, the main objective of this research was to explore the influence exerted by news reports about COVID-19 vaccines on people's sentiments, chronologically over one year. The method employed in this research achieved this objective by following similar

methods as those noted in Section 2, which focused on extracting tweets for sentiment analysis. In our study, the chronological analysis of UK-based COVID-19 vaccine-related tweets revealed how news events affected people's opinions. We adopted the BERT model because several studies had shown that BERT outperformed other models in tasks related to sentiment analysis. For example, in a study by Devlin et al. [34], BERT was trained on a large corpus of tweets and was able to achieve an accuracy of 85% in sentiment analysis, which outperformed other pre-trained models, such as Embeddings from Language Models (ELMO) and Generative Pretrained Transformer 2 (GPT-2), as well as traditional machine-learning models, such as Support Vector Machine (SVM) and logistic regression. BERT's pre-training on a large corpus ensured it was robust and less prone to over-fitting.

## 4. Results

In this section, we analysed the sentiments contained in tweets related to the three vaccines used in the UK. The periods of this study were based on *BBC News* reports regarding significant vaccine developments from January 2021 to December 2021.

### 4.1. Period 1: 25 January–1 February 2021

On 25 January 2021, the *BBC* reported that the Moderna COVID-19 vaccine appeared to work against more infectious variants of the virus, according to the US pharmaceutical company. Although this news was specifically about Moderna, this vaccine had only 65 tweets, as compared to the 632 tweets for AstraZeneca and 508 for Pfizer.

As shown in Figure 4, the sentiments were generally negative regarding the three vaccines. However, the percentages of negative tweets for AstraZeneca and Pfizer were above 50%. Moderna had more positive tweets than the others.

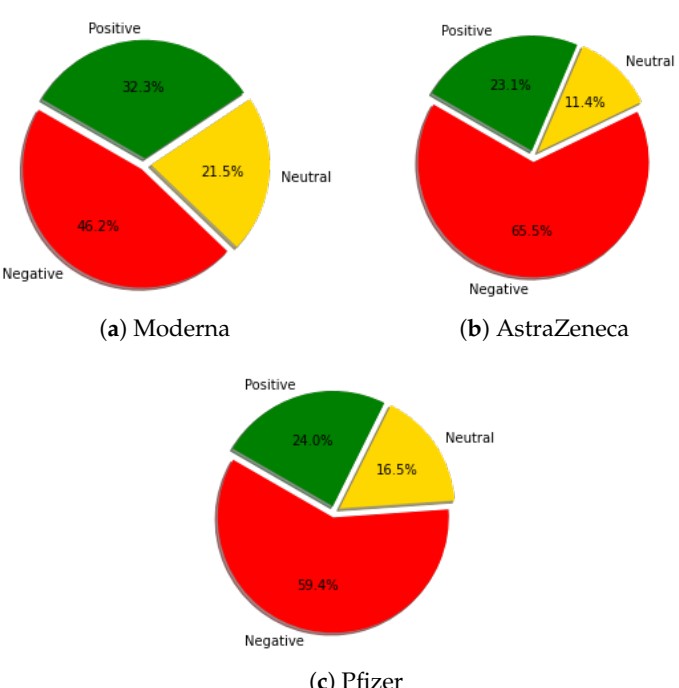

(**a**) Moderna      (**b**) AstraZeneca

(**c**) Pfizer

**Figure 4.** Period 1—Sentiment percentage of tweets per vaccine.

Figure 5 shows words that were used in the negative-oriented tweets. Although we can see some positive words such as *efficacy, good, and better*, there were numerous negative words in these tweets, as well. For Moderna, we found words such as *threatening, wrong, worse, delayed, and deficient*. Tweets related to AstraZeneca contained negative words such as *failed, threaten, issue, wrong, and dispute*. There are other negative sentiments concerning *profitability and nationalism*, in which vaccine companies were accused of being *profit*-focused

and the government accused of exhibiting *nationalism*. Tweets related to Pfizer had negative terms, as well, such as *risk, threatening, problem, and failed*.

Over this same period, some positive words were associated with the vaccines, as shown in Figure 6:

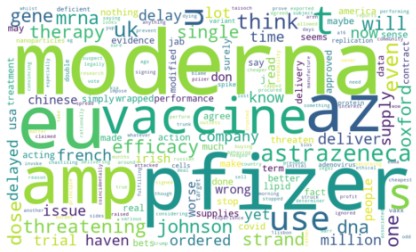

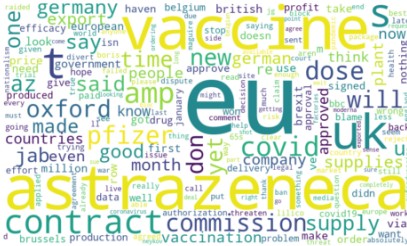

(**a**) Moderna

(**b**) AstraZeneca

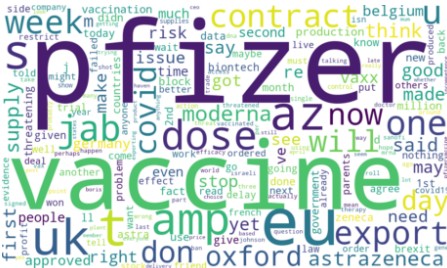

(**c**) Pfizer

**Figure 5.** Period 1—Words associated with negative tweets.

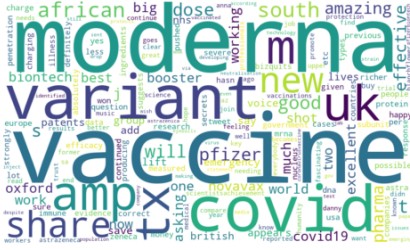

(**a**) Moderna

(**b**) AstraZeneca

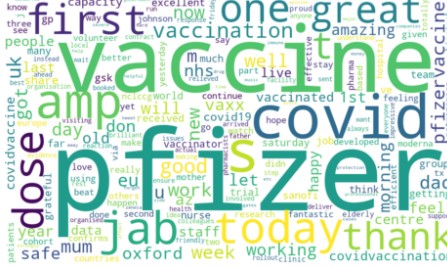

(**c**) Pfizer

**Figure 6.** Period 1—Words associated with positive tweets.

**Moderna**: *amazing, excellent, working, happy, best*.
**AstraZeneca**: *great, good, thank, amazing, interesting*.
**Pfizer**: *great, thank, happy, excellent, fantastic*.

*4.2. Period 2: 2 April—8 April 2021*

On 2 April 2021, the *BBC* reported 30 cases of blood clots in AstraZeneca recipients in the UK. During this period, the number of tweets for Moderna slightly increased from 65 to 92 while those for AstraZeneca and Pfizer decreased from 632 to 415 and from 508 to 230, respectively. This was surprising, as one would expect to see more tweets about AstraZeneca due to the reported blood-clot cases. However, the reason for the declining number of tweets about AstraZeneca could have been a result of the fading enthusiasm for the vaccine by the British populace, based on a survey of 5000 people, as reported by a *Reuters'* news article [36]. The article noted rising unease about the possible link to rare adverse side effects of the vaccine, but it also highlighted overall high confidence in vaccines in the UK. The article further reported that the AstraZeneca vaccine was suspended by over a dozen European countries due to reports of blood clots and low platelets in a small number of people who had received the vaccine.

Overall, for all the vaccines, the percentage of positive sentiments increased slightly while the percentage of negative sentiments decreased, as shown in Figure 7. However, the percentage of negative sentiments for Moderna increased slightly, from 46.2% to 46.7%, while positive sentiments increased from 32.3% to 34.8%. For AstraZeneca, the percentage of negative sentiments decreased from 65.5% to 57.1% while the percentage of positive sentiments increased from 23.1% to 30.1%, which was a significant increase for AstraZeneca. The percentage of negative sentiments for Pfizer decreased from 59.4% to 50.0% while the positive percentage increased from 24.0% to 35.2%. Although the sentiments were still generally negative during this period, there was a significant shift in sentiment from negative to positive.

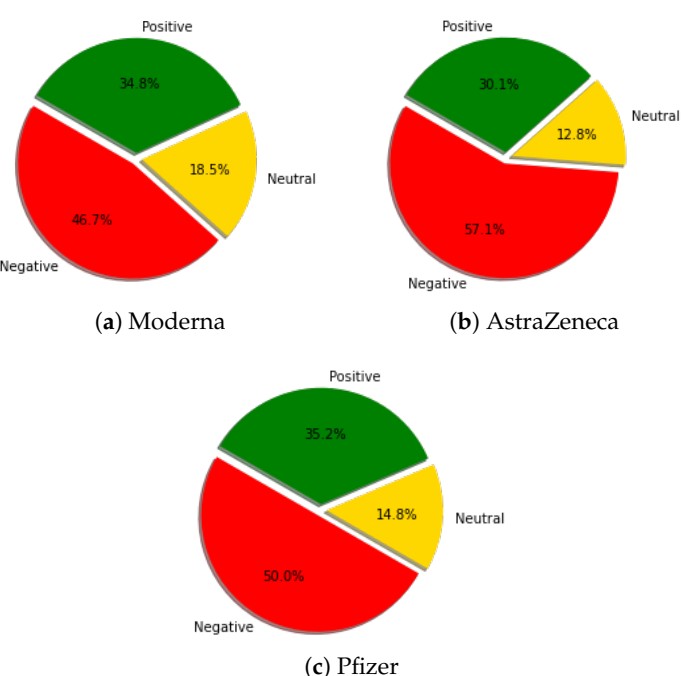

(**a**) Moderna      (**b**) AstraZeneca

(**c**) Pfizer

**Figure 7.** Period 2—Sentiment percentage of tweets per vaccine.

Figure 8 shows the word-clouds for negative words associated with the vaccines. During this time period, negative words associated with the vaccines included the following, with some overlap:

**Moderna**: *problem, clots, dead, effects, blood*.
**AstraZeneca**: *risk, severe, effect, clot, blood*.
**Pfizer**: *effect, death, blood, clot, problem*.

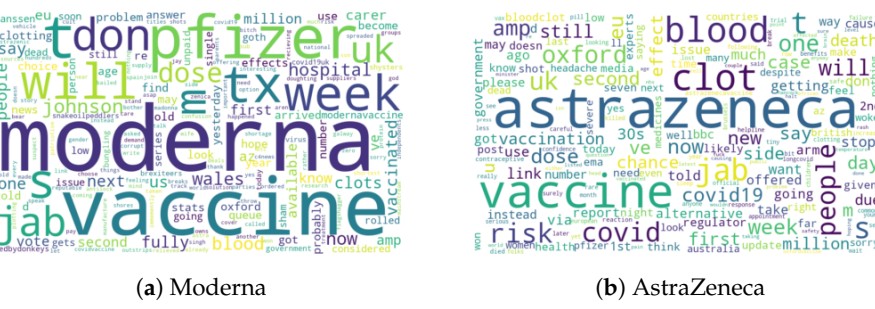

(**a**) Moderna    (**b**) AstraZeneca

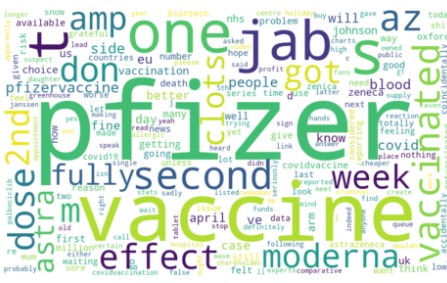

(**c**) Pfizer

**Figure 8.** Period 2—Words associated with negative tweets.

Over this same time period, positives words associated with the vaccines included the following (also see Figure 9):

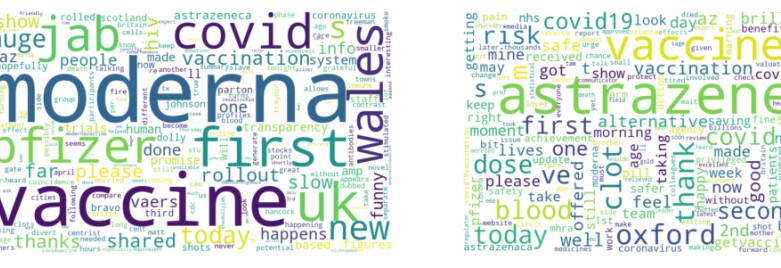

(**a**) Moderna    (**b**) AstraZeneca

(**c**) Pfizer

**Figure 9.** Period 2—Words associated with positive tweets.

**Moderna**: *huge, promise, transparency, funny, thanks.*
**AstraZeneca**: *good, brilliant, safe, happy, benefits.*
**Pfizer**: *thank, grateful, lovely, good, safe.*

*4.3. Time Periods 3–6*

Table 3 presents a summary of sentiment percentages for time periods 3–6. Generally, the overall sentiments for each of the vaccines were negative. The sentiment percentages were not indicative of the number of tweets about the vaccines. The table shows that

Moderna received fewer negative sentiments, as compared to AstraZeneca and Pfizer. It was also observed that AstraZeneca attracted the most negative tweets. Regarding positive sentiments, Moderna attracted more positive tweets than AstraZeneca and Pfizer, and Pfizer received more positive tweets than AstraZeneca. For all the time periods, Period 3 (23 May–29 May 2021) with the breaking news, 'COVID: Pfizer and AstraZeneca jabs work against Indian variant', received the least negative tweets and the most positive tweets. This was a significant shift in sentiment from Period 2 when 30 blood clots cases had been found in AstraZeneca recipients. The positive news reported in Period 3 resulted in Moderna's percentage of negative sentiment decreasing from 46.7% to 29.4%, AstraZeneca's percentage from 57.1% to 45.0%, and Pfizer's percentage from 50.0% to 43.7%. On the positive side, Moderna's percentage increased from 34.8% to 55.9%, AstraZeneca's percentage from 30.1% to 43.0%, and Pfizer's percentage from 35.2% to 43.4%. These percentages suggested that the news outbreaks could have impacted people's sentiments towards the vaccines.

Period 4 indicated another significant shift in sentiments but, this time, to the negative side. During this period (9 July–15 July 2021), *BBC News* reported the headline 'Heart inflammation link to Pfizer and Moderna jabs'. During this period, the percentage of negative sentiments for Moderna increased from 29.4% to 42.0%, AstraZeneca from 45.0% to 57.7%, and Pfizer from 43.7% to 51.8%. The percentages of positive sentiments for the three vaccines also decreased from 55.9% to 40.6%, from 43.0% to 26.9%, and from 43.4% to 38.2% for Moderna, AstraZeneca, and Pfizer, respectively. There were no significant changes for Periods 5 and 6. In general, people's sentiments appeared to shift in relation to breaking news.

**Table 3.** Periods 3–6: Sentiment percentage of tweets per vaccine.

| Timeline | Negative (%) | | | Positive (%) | | |
|---|---|---|---|---|---|---|
| | Moderna | AstraZeneca | Pfizer | Moderna | AstraZeneca | Pfizer |
| 23 May–29 May 2021 | 29.4% | 45.0% | 43.7% | 55.9% | 43.0% | 43.4% |
| 9 July–15 July 2021 | 42.0% | 57.7% | 51.8% | 40.6% | 26.9% | 38.2% |
| 26 Nov–2 Dec 2021 | 45.3% | 57.9% | 57.9% | 35.3% | 36.8% | 29.8% |
| 8 Dec–14 Dec 2021 | 42.0% | 57.6% | 52.4% | 36.2% | 24.2% | 29.3 |

## 5. Discussion

In summary, this paper contributes to the knowledge by exploring the general sentiment towards COVID-19 vaccines in the UK by chronologically examining the potential effects of news reports on the public sentiment towards COVID-19 vaccines and by comparing the sentiments toward three different COVID-19 vaccines in conjunction with these news reports.

In Section 2, we compared the contributions of existing relevant studies and the result of the proposed study. The study in [37] revealed that public opinion about COVID-19 vaccinations shifted dramatically over time and across locations. Sentiment analysis assisted public health authorities in building locally tailored vaccination education initiatives by providing timely insights into public sentiment concerning COVID-19 vaccines. However, the study only focused on identifying people's sentiments and did not consider how external factors, such as news reports, affected people's views. In Rahul et al. [38], they examined sentiment analysis and topic modelling of COVID-19 vaccine-related tweets from 1 November 2020 to 16 December 2020. The study reported more positive than negative sentiments after analysing more than 500,000 tweets. In [39], the authors conducted a study to determine the influence of COVID-19 news on Twitter sentiment in four different countries: the UK, India, Japan, and South Korea. The UK was found to have one of the highest percentages of negative sentiment. Surprisingly, the UK was the most affected among the four countries. This study, however, only focused on analysing news headlines without the corresponding impact on people's sentiments, especially towards COVID-19

vaccines. Gesualdo et al. [23] posed the question, "How do Twitter users react to TV broadcasts dedicated to vaccines in Italy?" They revealed that social media discussions in response to public news had a significant impact priceless and could be used to inform vaccine promotion communication strategies. While this study was similar to ours, it did not apply a chronological strategy to its analysis. Furthermore, the study focused on the Italian population; therefore, it was also important to conduct a study focused on the UK.

While our study makes important contributions to the knowledge and the theory, it also provided valuable insights for policymakers and practitioners in several ways:

- Public sentiment towards vaccines: The findings provide policymakers and practitioners with an understanding of public sentiment towards COVID-19 vaccines in the UK, which could assist them in developing more effective communication strategies to increase vaccine uptake.
- Impact of news and events on public sentiment: The findings provide policymakers and practitioners with an understanding of how news and events impact public sentiment towards COVID-19 vaccines. This could assist with developing more effective communication strategies to address any misconceptions or concerns that people may have about vaccines.
- Misinformation about vaccines: The findings provide policymakers and practitioners with an understanding of the extent of misinformation about COVID-19 vaccines on social media, which could assist in developing strategies to counter misinformation and protect public health.
- Targeted interventions: The findings provide policymakers and practitioners with insights into vaccine hesitancy and could assist in developing targeted interventions to increase vaccine uptake.
- Development of Communication strategies: The findings provide policymakers and practitioners with insights on how best to communicate the vaccine message to the public, including the ideal timing and channels for optimal impact.

## 6. Conclusions

COVID-19 vaccines are important for global recovery after the COVID-19 pandemic. However, many people have been sceptical about the available vaccines. Vaccine hesitancy is a global problem, and it is important to understand the reasons for this in order to address them. This study aimed to understand people's sentiments towards the COVID-19 vaccines in the UK by examining *BBC News* reports. In particular, we analysed tweets in relation to three COVID-19 vaccines—Moderna, AstraZeneca, and Pfizer—over a twelve-month period according to data from six chronological time periods. We found that, overall, the Moderna vaccine was generally perceived more positively than the other two vaccines while the AstraZeneca vaccine attracted more negative sentiments. These perceptions were largely due to the slightly lower risk of Moderna vaccines, as compared to the others, and the major blood clot concerns of the AstraZeneca vaccine, respectively.

The study achieved the objectives of the research. It explored the general sentiments towards COVID-19 vaccines among the UK population. We found that the general sentiment about vaccines during the study periods was negative. A high vaccination rate was originally thought to be well received, and it could potentially overcome vaccine hesitancy, regardless of the fears and uncertainties about the pandemic. However, we demonstrated that this was not the case, as various breaking news, whether positive or negative, appeared to sway people's opinions. The news stories, as reported by the *BBC*, were based on developments concerning vaccines during the pandemic.

For all six chronological periods selected for this study, Period 1 (25 January–31 January 2021) with the breaking news, 'Moderna vaccine appears to act against variations' had the highest record of associated negative sentiments. Period 3 (23 May–29 May 2021) was associated with the fewest negative tweets and the most positive tweets, and it has the breaking news, 'COVID: Pfizer and AstraZeneca jabs effective against Indian variant'. The study showed that there was a relationship between the news and people's perceptions of

the COVID-19 vaccines. The results indicated that news had a significant impact on how people perceived the vaccines, as indicated by the shifts in sentiment percentages. This suggested that the media played a role in shaping public opinion on vaccines.

In order to address vaccine hesitancy, it is therefore important to provide accurate and balanced information about vaccines in the media. Based on the result of this study, there are several things that should be considered in order to manage the public sentiment toward COVID-19 vaccines (and other vaccines) and vaccine hesitancy. First, it is important to provide accurate and updated information about vaccines. Second, it is important to address the concerns of the people. The findings of this study could mean various things to various stakeholders. For governments and policymakers, the understanding that news publications affect people's opinions about events, in this case, COVID-19 vaccines, could lead to the implementation of policies that would enable them to monitor the information being conveyed about the events and how people are reacting to them. It could also assist in tracking vaccine hesitancy and the themes that influence the hesitation in order to implement or adopt better strategies in addressing people's hesitancy. An effective and authoritative information dissemination process would need to be adopted in promoting and strengthening vaccine confidence. Advancing vaccine confidence could be accomplished through sustainable engagements with the general public and partnerships with local authorities, community leaders, influencers, and cultural organisations, in an effort to promote COVID-19 vaccines. The government would also need to stimulate more responsive, equitable, and accessible programs that reduce barriers to vaccine uptake and increase confidence in vaccines. Healthcare practitioners are the most trusted influencers and advisors of vaccination decisions. They influence people's opinions and sentiments about vaccines; however, they have been faced with erratic changes in people's opinions and sentiments, largely due to news reportage and concerns about vaccines. An understanding of how news reportage sways people's opinions and affects vaccine hesitancy could enable them to implement strategies to manage the public's shifting views.

It is recommended that future studies be conducted to investigate the impact of the news on other aspects of people's lives, such as their emotions and economic decisions. In addition, future studies could also explore the impact of misinformation (i.e., 'fake news') on people's sentiments and behaviours.

This study had limitations as it only focused on tweets generated in the UK. In addition, the study only used a small sample of tweets as the data were collected over short periods of time. Future studies could focus on covering more timelines and include tweets from other countries to understand how social media users perceive COVID-19 vaccines in different countries.

The *BBC* has been used as a source of information in this research because of its wide reach and broad impact. The *BBC* has a large audience, and its content is widely shared and distributed through various platforms. Due to its reach and reputation, the *BBC*'s communicative power was considered relevant. However, we acknowledge that there were other sources of influence including social media platforms, other newspapers, radio, and television. Additionally, individuals are also influenced by their personal experiences, friends, family, and other people in their social networks. Therefore, it may be more accurate to state that among the many sources of information available to individuals, the *BBC* continues to be highly relevant worldwide.

**Author Contributions:** Conceptualization by R.F. and E.I.; Methodology by O.A.; Software by E.I. and O.A.; formal analysis by U.O. and C.I.; Investigation by E.I. and C.I.; Resources and data collection by O.A., R.F. and E.I.; Writing by: R.F., O.A. and U.O.; Validation by: U.O. and C.I. All authors have read and agreed to the published version of the manuscript.

**Funding:** This research received no external funding.

**Institutional Review Board Statement:** Not applicable.

**Informed Consent Statement:** Not applicable.

**Data Availability Statement:** Data can be accessed via Twitter API.

**Conflicts of Interest:** The authors declare no conflict of interest.

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
