# Peer review of "Sentiment Computation of UK-Originated COVID-19 Vaccine Tweets: A Chronological Analysis and News Effect"

_sustainability, doi:10.3390/su15043212_

Round 1
Reviewer 1 Report (New Reviewer)
The article is fluent, linear, and adequately supported. The clear methodology leads precisely to answering the research questions. However, at a time characterized by a total bubble of information reaching individuals from official and less official sources summarising the mechanism of influence on the sentiment of tweets to the communicative power of the BBC alone is in my opinion an extreme simplification that can run into easy and obvious spurious reports. The only suggestion I would like to make in this regard is to soften the directionality of the influence relationship and to clarify that among the many sources of information that can reach the subjects, the BBC has a relevant weight due to its diffusive characteristics, aware of the fact that it is not the only isolable source of influence.
Author Response
Kindly see the attached response.

Reviewer 2 Report (New Reviewer)
The study adressed a very interesting concern about the british society sentiment over some major public health actions - vaccines against COVID-19 virus. The introduction is comprehensive and sufficient, while the methodology part is very well elaborated. The results show how general sentiment over three types of vaccines have been changing over six periods of time, in the UK, while the representation of the results is eficient.
I have found few comments to address the authors:
1. The text betweeen lines 110 to 202, the Literature review can be moved after the results section and disscussed in comparison with the findings of the present study.
2. At line 213, the phrase "As earlier mentioned, the dataset used in this study is extracted from Twitter over a 213 period." is in contradiction with the one next to it. I recommend erasing it.
3. Line 311: Figure 8. Period 1 - Words associated with Negative tweets - replace with Period 2.
Author Response
Kindly see the attached response.

Reviewer 3 Report (Previous Reviewer 3)
The introduction section needs significant improvement. Please see my concerns below.
The main objectives of this study should be better discussed at the beginning of the introduction section rather than leaving it to the end of the introduction section.
The authors argued that the contribution of this paper is 4-fold:
· To explore the general sentiment about COVID-19 vaccines in the UK
· To chronologically examines the potential effects of news reports on the public sentiment towards COVID-19 vaccines
· To compare the sentiments toward different COVID-19 vaccines in line with these news reports
· To compare the sentiment towards COVID-19 vaccines over time
But these “contributions” read like the research questions the study aims to address. You need to clarify, for example, what are the research gaps this study aims to address? How and why addressing the gap advances knowledge, and how the research findings provide insights for policymakers and/or practitioners.
More detailed information about research methodology should be provided, for example, to better discuss why the methods used in this study can answer the research questions shown in the introduction section.
To better discuss the main findings of this study, a separate section on discussion should be provided. In this section, you could discuss the findings in more detail for example by comparing your results with that of previous research (see the work you listed in the literature review section), and clarify how your research findings advance knowledge and contribute to theory. You might also include one section to clarify what insights your research findings can provide for practitioners and/or policymakers, can they learn something new from your research?
Author Response
Kindly see the attached response.

Round 2
Reviewer 3 Report (Previous Reviewer 3)
Thanks for revising the manuscript.
This manuscript is a resubmission of an earlier submission. The following is a list of the peer review reports and author responses from that submission.
Round 1
Reviewer 1 Report
Recommendation: Major revision.
Detailed review comments:
This manuscript explored the impact of news reports on COVID-19 vaccines on the public sentiments about the vaccines. The paper collected tweets about different types of vaccines (Pfizer, AstraZeneca, Moderna) from UK within one week after each of the six BBC news report on the vaccines, and carried out sentiment analysis and word frequency analysis. The results showed that the BBC News reports on COVID-19 vaccines did significantly impact public sentiments regarding the COVID-19 Vaccines.
I have the following concerns.
1. The paper focused too much on empirical discussion, therefore suffered from lack of theoretical relevance. As far as I know, many scholars have examined the relationship between media information disclosure and public sentiment.
2. The contribution statement (Line 94-100) does not clearly show the novelty of the paper.
3. The paper obtained some interesting findings (for example, the number of tweets related to the AstraZeneca vaccine has not increased yet decreased after the negative news reports about the vaccine mentioned in section 4.2). However, the paper did not explore the reasons in depth, but only concluded that "news reports significantly impact the public sentiment", which making the research superficial.
4. The title of the manuscript did not show the research question and the key independent variable of this study.
5. How does the paper relate to sustainability?
Author Response
Many thanks for your review. Attached is the point-to-point response to your comments. Thanks.

Reviewer 2 Report
INCORPORATES THE NECESSARY INFORMATION, GUIDING THE READER TO IDENTIFY THE BASIC CONTENT OF THE TEXT QUICKLY AND TO DETERMINE ITS RELEVANCE. IS SEMANTICALLY SELF-SUFFICIENT. MAKES EXPLICIT THE PURPOSE AND AIMS OF THE TEXT PRESENTED.
THE TITLE SUMMARISES THE MAIN IDEA OF THE TEXT, IT IS SELF-EXPLANATORY. IT IS ALSO CONCISE AND INFORMATIVE.
PRESENTS THE OBJECTIVE OF THE STUDY, THE MAIN ELEMENTS OF THE METHODOLOGY AND THE FINDINGS AND CONCLUSIONS.
IT IS A CURRENT AND THEMATICALLY RELEVANT ARTICLE.
AS FOR THE RESULTS, ANALYSIS AND DISCUSSION, THEY ARE CORRECTLY PRESENTED, TAKING UP THE METHODOLOGICAL PROCEDURES AND CONCEPTUAL GUIDELINES FOR THE ANALYSIS OF RESULTS. CONCLUSIONS of interest appear, which are based on the powerful statistical analysis.
Author Response
Many thanks for your review. We appreciate your kind reviews.
Reviewer 3 Report
This study talks about an interesting topic, but suffers from several specific limitations, which I outline below.
In the introduction section, the authors need to discuss the main research objectives / questions more clearly. The authors listed several potential contributions, but a further clarification is needed here in order to better discuss how the findings of this research can make such contributions. Also, this should be further strengthened in the discussion section.
The literature review section looks very thin. A comprehensive literature review is needed, also to be better linked with the objectives of this research.
The results of the sentiment analysis are generally well reported, but not well discussed. Let me be more specific here.
The conclusion/discussion section needs significant improvement. As noted above, the authors could use this section to better discuss how the empirical findings make contributions listed in the introduction section. But at its current presentation, the discussion looks very descriptive, and mainly summarizes the results of the sentiment analysis without any in-depth analysis.
The authors could also discuss how the empirical findings provide practical implications for practitioners and/or policymakers, which will help to strengthen the main contributions of this research.
Author Response

(The authors gave the same response as above.)

Round 2
Reviewer 1 Report
Thank you for replying to my questions and making the revisions.
The authors addressed my questions, except the first point on theoretical relevance. The paper did not engage with the literature on the theoretical discussion on the potential effect of news on public sentiments.
I think the paper may have some merits, and may be suitable for a slightly lower tier journal.
Reviewer 3 Report
I appreciate the improvements that the authors have made and I feel that the paper reads better now.